# Identification of Friction Behavior Variation in the Minor Flank of Square Shoulder Milling Cutters under Vibration

**Bin Jiang, Weiheng Li, Peiyi Zhao \*, Lili Fan and Meng Sun**

Key Laboratory of Advanced Manufacturing and Intelligent Technology, Ministry of Education,
Harbin University of Science and Technology, Harbin 150080, China; jiangbin943@hrbust.edu.cn (B.J.);
liweiheng2021@163.com (W.L.); 1910100015@stu.hrbust.edu.cn (L.F.); sunmeng7057@163.com (M.S.)
\* Correspondence: zhaopeiyi@hrbust.edu.cn

**Abstract:** In the milling process, the friction and wear of the tooth minor flank of the square shoulder milling cutter directly affects the machined surface quality and the cutter's life. The friction of the minor flank of the cutter tooth presents a nonlinear distribution, and its variation cannot be revealed by using a single parameter. It is difficult to identify the dynamic characteristics of the friction of the minor flank of the cutter tooth. In this work, the friction velocity model for the cutter tooth minor flank was developed by using the relative motion relationship between the flank area element and the workpiece transition surface. In accordance with the atomic excitation theory developed under the potential energy field at the friction interface of the cutter, the model for friction energy consumption under the friction velocity and thermal-stress coupling field on the minor flank of the cutter tooth was developed. Based on the mechanism of the interfacial atomic thermal vibration, the model for the friction coefficient under thermal-stress mechanical coupling was developed. Using the instantaneous friction coefficient and normal stress, the instantaneous friction distribution function of the flank was obtained. Finally, an identification method for the friction dynamic characteristics of the shoulder milling cutter tooth flank under vibration was proposed and verified by experiments.

**Keywords:** square shoulder milling cutter; minor flank of cutter tooth; friction speed; friction energy consumption; friction coefficient





## 1. Introduction

The friction and wear of the minor cutting edge of the tooth of the square shoulder milling cutter directly affects the machined surface quality of the workpiece. The friction properties of the minor flank of the tooth determine the wear status and cutting performance of the milling cutter. The effective identification of the friction speed, energy consumption, friction coefficient and friction force of the minor flank are important indexes for revealing the variation in the friction and wear status of the cutter-workpiece [1,2]. During the high speed and intermittent cutting process, the instantaneous contact relationship between the minor flank of the cutter tooth and the workpiece changes continuously. This is affected by the impact of cutter teeth cutting into and out of the workpiece and the milling vibration. The friction speed, friction energy consumption, friction coefficient and friction force on the minor flank of the cutter tooth were present complex dynamic changes. It is difficult to realize the accurate control of the tool's wear and the machined surface quality of the workpiece [3,4]. Therefore, it is necessary to establish the correct models and methods for the friction coefficient and friction force of the minor flank of the cutter tooth and reveal its dynamic characteristics. It is of great significance to inhibit cutter wear and improve cutter life [5].

The instantaneous multi-tooth cutting method of a square shoulder milling cutter determined the forming of the machined surface of the workpiece. The instantaneous contact relationship between the minor flank of the cutter tooth and the machined transition surface of the workpiece was the key to revealing the dynamic characteristics of the friction

variables [6–8]. The contact between the minor flank of the cutter tooth and the machining transition surface of the workpiece was in an unstable state under the milling vibration. This resulted in constant variation in the friction variables, which made the friction dynamic characteristics of the minor flank of the cutter tooth uncertain [9–12].

At present, the friction and wear of the cutter tooth's minor flank for square shoulder milling cutters mainly focuses on the detection. Based on the cutting force model and wavelet packet decomposition, Zhang X [13] proposed a deep-learning-based method for monitoring tool wear in the milling of complex parts using pre-selected features. Sousa V [14] used multiple machining experiments to measure the cutting edge wear, revealing the tooth wear mechanism and its formation process. An Q [15] established the theoretical model of flank wear width. The above scholars have made great contributions to tool wear monitoring and theoretical model construction, but they focus on the cutting mechanics caused by the wear of the main cutting edge and its flank of the milling cutter and the structural response of the cutting edge and the flank. The variability of the friction and wear process of the minor flank of the cutter under the action of milling vibration is ignored. The dynamic characteristics of the flank friction of the cutter tooth have yet to be revealed. It is necessary to research this on the basis of existing research.

Establishing correct models for the friction speed, friction coefficient, friction energy consumption and friction force is the key to reveal the dynamic characteristics of minor flank friction of a cutter tooth. Gao H [16] performed intermittent machining of hardened steels with uncoated tools to determine the cutting performance and wear/damage characteristics of the tools. Mntyl A [17] proposed a new COF and wear evolution model based on the COF evolution law driven by friction energy dissipation and on the classical Archard equation as the evolution law of wear depth. Kitamura T [18] used the friction coefficient of the micro feed end cutting force to evaluate fluid milling. Zhang C [19] established a new local friction law. The above models and methods have important guiding significance for constructing friction speed, friction energy consumption, friction coefficient and friction force models. However, they ignored the variability of the friction velocity vector on the minor flank of the milling cutter tooth and the time variability of the instantaneous contact stress and contact temperature between the minor flank of the milling cutter tooth and the machining transition surface. There is a shortage in the dynamic behavior of the flank friction characteristic variables of the cutter tooth under vibration.

In this paper, the model of the milling cutter posture under vibration was developed. The instantaneous contact relationship between the minor flank of the cutter tooth and the machining transition surface of the workpiece under vibration was studied. The identification method of the upper and lower wear boundaries of the minor flank was proposed. The time-varying characteristics of the wear boundary of the minor flank were unveiled. According to the interface atomic potential energy theory, the models of the instantaneous friction velocity, energy consumption, friction coefficient and friction force were developed. Finally, an identification method for the friction characteristic variable of the minor flank of the cutter tooth was proposed. The effectiveness of the method was verified by experiments.

## 2. Instantaneous Posture of the Milling Cutter and the Minor Cutting Edge

A square shoulder milling cutter is a typically used tool in the machining of titanium alloy structure parts. A three-blade square shoulder milling cutter and its structure provided by the cooperative enterprise for efficient cutting of titanium alloy structural parts are shown in Figures 1 and 2. The variable's interpretation is shown in Table 1.

**Table 1.** Explanation of the cutting behavior of the cutter under vibration.

| Serial Numbers | Parameters | Parameters Explanation |
|---|---|---|
| 1 | $r_{max}$ | Maximum radius of gyration at the tip of the cutter tooth. |
| 2 | $r_i$ | The rotation radius of the $i$th cutter tooth. |
| 3 | $D$ | The diameter of the milling cutter. |
| 4 | $\Delta r_i$ | The radial error of the $i$th cutter tooth. |
| 5 | $\Delta Z_i$ | The axial error of $i$th cutter tooth. |
| 6 | $l_i$ | Distance from the lowest point of the $i$th cutter tooth to the end face of the milling cutter. |
| 7 | $l_1$ | The total length of the milling cutter. |
| 8 | $\eta_{ii+1}$ | The angle between the $i$th cutter tooth and the $(i + 1)$th cutter tooth. |
| 9 | $O_i\text{-}X_iY_iZ_i$ | The cutter tooth coordinate system: $O_i$ is the tip point of the $i$th cutter tooth; $X_i$ passes the minor cutting edge and points to the center of the milling cutter; $Y_i$ is perpendicular to the instantaneous cutting speed direction; $Z_i$ is perpendicular to the $X_i$ axis and $Y_i$ axis. |
| 10 | $O_0\text{-}X_0Y_0Z_0$ | The milling cutter structure coordinate system: $O_0$ is the center of rotation of the lowest cutter tip in the axial direction; $X_0$ is over $O_0$ and perpendicular to $Y_0$; $Y_0$ crosses the center point $O_0$ and points to the tool tip point of the maximum outer diameter circle; $Z_0$ is perpendicular to the $X_0$ axis and $Y_0$ axis. |
| 11 | $ds$ | Surface element. |
| 12 | $dX_i, dY_i$ | The length of the surface element in $X_i$ axis and $Y_i$ axis directions. |
| 13 | $l_{X1}, l_{X2}, l_{X3}$ | Length of three segments of the minor cutting edge. |
| 14 | $l_\gamma$ | Direction vector of the surface element. |
| 18 | $p_s$ | Cutting plane. |
| 19 | $\lambda'_s$ | End of the dip angle of the blade. |
| 20 | $k'_r$ | End of the cutting edge angle |
| 21 | $\alpha'_0$ | End of the relief angle. |
| 22 | $l_{Y_1}$ | The intersecting line of the perpendicularity plane and the cutting plane. |
| 23 | $A_i(X_i,Y_i,Z_i)$ | Minor flank. |
| 24 | $o\text{-}xyz$ | Workpiece coordinate. $x, y, z$ are the length, width, height direction of the workpiece. |
| 25 | $o_0\text{-}uvw$ | Milling cutter cutting coordinate without vibration. $u, v, w$ parallel to the length, width and height direction of the workpiece, respectively |
| 26 | $O_c\text{-}UVW$ | Milling cutter cutting coordinate with vibration. $U, V, W$ parallel to $X, Y, Z$ axis in the initial state, rotate around the coordinate system $Os\text{-}XYZ$ when milling. |
| 27 | $a_p$ | The cutting depth of the milling cutter. |
| 28 | $a_e$ | The cutting width of the milling cutter. |
| 29 | $n$ | The rotation speed of the milling cutter. |
| 30 | $v_f$ | The nominal feed speed of the milling cutter. |
| 31 | $\theta(t)$ | Instantaneous attitude angle of the milling cutter under vibration. |
| 32 | $\varphi_i$ | The instantaneous position angle of the cutter teeth in the cutting period determined by the milling cutter speed |
| 33 | $\varphi_g(t_0)$ | The initial cutting angle of the milling cutter. |
| 34 | $\varphi_q(t_0)$ | The angle between structural coordinate system of the cutter and the cutting motion coordinate system under vibration at time $t_0$. |

| Serial Numbers | Parameters | Parameters Explanation |
|---|---|---|
| 35 | $A_x(t)$, $A_y(t)$, $A_z(t)$ | The vibration displacement of the milling cutter in three directions of $x$, $y$ and $z$, respectively. |
| 36 | $l$ | Milling cutter overhang. |
| 37 | $\theta_1(t)$ | The projection angle of the milling cutter attitude angle $\theta(t)$ on the $vo_0w$ plane. |
| 38 | $\theta_2(t)$ | The projection angle of the milling cutter attitude angle $\theta(t)$ on the $uo_0w$ plane. |
| 39 | $\theta_{mX}$ | Angle between projection of $X_i$ axis on $xoy$ plane and $x$ axis. |
| 40 | $\theta_{mY}$ | Angle between $Y_i$ axis and $xoy$ plane. |
| 41 | $\theta_{mZ}$ | The angle between projection of $X_i$ axis and $X_i$ axis on $xoy$ plane. |
| 42 | $\theta_{my}$ | Angle between friction velocity $v_m$ and $Y_i$ axis. |
| 43 | $\theta_{mz}$ | Angle between $v_m$ and $Z_i$ axis. |
| 44 | $\theta_m$ | The angle between $v_m$ and relative motion velocity $v_n$. |
| 45 | $\theta_c$ | Angle between $vn$ and vector $l_p$. |
| 46 | $\varphi_i'(t_0)$ | The position of the tip point of the *i*th tooth of the milling cutter at the initial cut-in moment during milling |
| 47 | $l_p$ | Projection vector of $v_n$ on the common tangent plane of the cutter minor flank and workpiece transition surface. |
| 48 | $o_r$ | Center point of the surface element |
| 49 | $l_u$ | Friction upper boundary |
| 50 | $l_d$ | Friction lower boundary |
| 51 | $\mu$ | Friction coefficient |
| 52 | $f_p$ | Friction force |
| 53 | $v_m$ | Friction speed |

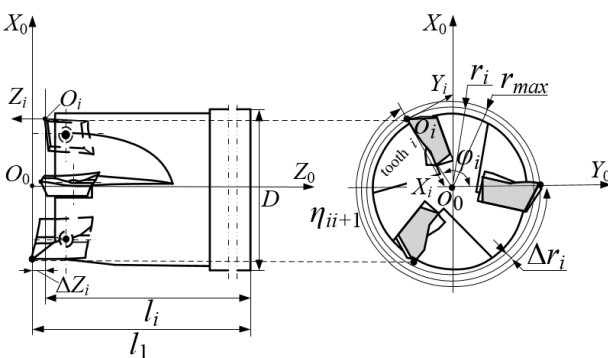

**Figure 1.** The structure of the square shoulder milling cutter.

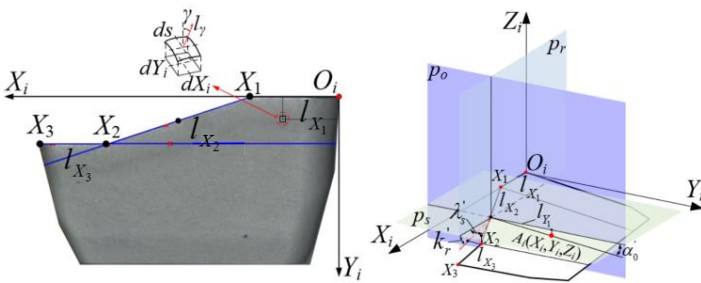

**Figure 2.** Minor cutting edge and minor flank of the cutter teeth.

$A_i$ is the intersection equation of the minor flank and $p_0$ plane, which could be expressed as:

$$\frac{\partial A_i(X_i, Y_i, Z_i)}{\partial Y_i} = \tan \alpha'_0 \tag{1}$$

$$A_i(X_i, Y_i, Z_i) = \begin{cases} X_i \\ Y_i = (X_i - l_{X_1}) \cdot \tan \lambda'_s + l_{Y_1} \\ Z_i = (X_i - l_{X_1}) \cdot \tan k'_r + l_{Y_1} \cdot \tan \alpha'_0 \end{cases} \tag{2}$$

When $\lambda'_s = 0$ and $k'_r = 0$, the minor cutting edge $l_{X_1}$ could be expressed as:

$$l_{X_1}(X_i, Y_i, Z_i) = \begin{cases} X_i \\ Y_i = 0 \\ Z_i = 0 \end{cases} \tag{3}$$

When $\lambda'_s$ and $k'_r$ are not 0, the minor cutting edge $l_{X_2}$ could be expressed as:

$$l_{X_2}(X_i, Y_i, Z_i) = \begin{cases} X_i \\ Y_i = (X_i - l_{X_1}) \cdot \tan \lambda'_s \\ Z_i = -(X_i - l_{X_1}) \cdot \tan k'_r \end{cases} \tag{4}$$

The minor cutting edge equation $l_{X_3}$ of the cutter tooth could be derived as:

$$l_{X_3}(X_i, Y_i, Z_i) = \begin{cases} X_i \\ Y_i = l_{X_2} \cdot \tan \lambda'_s \\ Z_i = -l_{X_2} \cdot \tan k'_r \end{cases} \tag{5}$$

The cutting trajectory of the milling cutter and cutter tooth were characterized by the motion of the milling cutter coordinate origin and the cutter tooth coordinate origin relative to the workpiece under vibration. The instantaneous cutting posture angle of the cutter and its tooth was characterized by the cone angle formed by the axis of the cutter and its tooth and the axis of the cutter at the predetermined position, as shown in Figure 3.

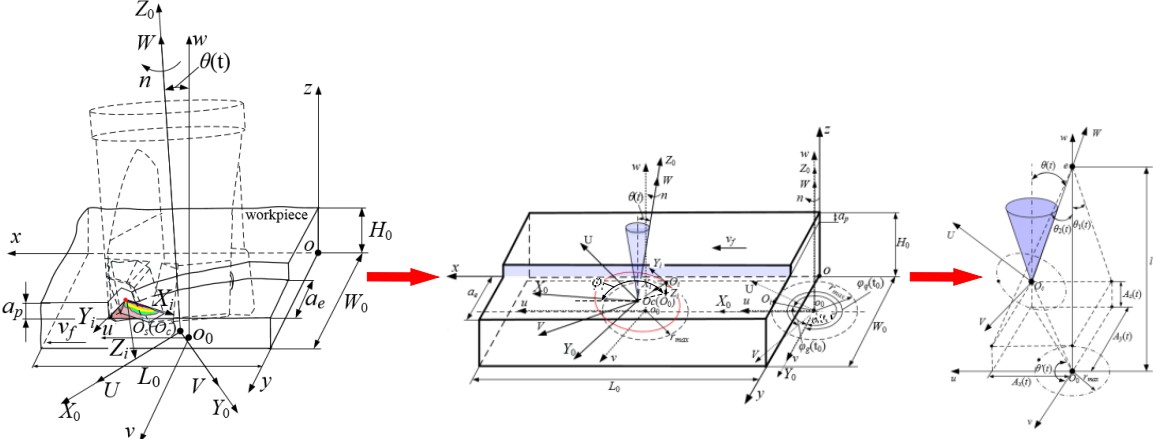

**Figure 3.** Motion state of the milling cutter and the cutter tooth under vibration.

In the workpiece coordinate system, the trajectory equation of any point on the minor cutting edge could be expressed as:

$$\begin{bmatrix} x & y & z & 1 \end{bmatrix}^T = \vartheta \cdot \begin{bmatrix} X_i & Y_i & Z_i & 1 \end{bmatrix}^T \tag{6}$$

where $\vartheta$ is the conversion relationship between the cutter tooth coordinate system and the workpiece coordinate system.

The angle $\varphi_q(t)$ between the $V$ axis and $Y$ axis in the $UVW$ plane could be expressed as:

$$\varphi_q(t) = \varphi_g(t_0) + \varphi_i + 2\pi n(t - t_0) - \left\lfloor \frac{\varphi_g(t_0) + \varphi_i + 2\pi n(t - t_0)}{2\pi} \right\rfloor \cdot 2\pi \tag{7}$$

In the equation, $\varphi_g(t_0)$ could be expressed as:

$$\varphi_g(t_0) = \arccos\left(\frac{a_e - r_{\max}}{r_{\max}}\right) \tag{8}$$

The instantaneous position coordinates $(x_{o0}(t), y_{o0}(t), z_{o0}(t))$ of the coordinate origin $o_o$ of the milling cutter cutting coordinate system in the workpiece coordinate system $o\text{-}xyz$ without vibration could be derived as:

$$\begin{cases} x_{o0} = v_f t - r_{\max} \\ y_{o0} = W_0 - a_e \\ z_{o0} = H_0 - a_p \end{cases} \tag{9}$$

The instantaneous angle $\theta_1(t)$ between the projection of the $W$ axis on the $vo_0w$ plane and the $w$ axis and the instantaneous angle $\theta_2(t)$ between the projection of the $W$ axis on the $uo_0w$ plane and the $w$ axis could be written as:

$$\theta_1(t) = \arctan\left(\frac{A_y(t)}{l - A_z(t)}\right), \theta_2(t) = \arctan\left(\frac{A_x(t)}{l - A_z(t)}\right) \tag{10}$$

The instantaneous offset angle $\theta(t)$ of the milling cutter cutting coordinate system with vibration could be calculated as:

$$\theta(t) = \arctan\frac{\sqrt{A^2{}_x(t) + A^2{}_y(t)}}{l - A_z(t)} \tag{11}$$

The titanium alloy milling experiment was carried out by using the three-tooth carbide shoulder milling cutter with a diameter of 25 mm provided by the cooperative enterprise. The milling experiment was carried out on a three-axis milling machining center (VDL-1000E).The cutting stroke was 2.5 m. The size of the titanium alloy specimen was 250 mm × 100 mm × 20 mm. Its material composition is shown in Table 2.

**Table 2.** Titanium alloy workpiece material composition.

| Element | Al | V | Fe | O | Si | C | N | H | Others | Ti |
|---|---|---|---|---|---|---|---|---|---|---|
| Contents (%) | 5.5~6.8 | 3.5~4.5 | 0.3 | 0.2 | 0.15 | 0.1 | 0.05 | 0.01 | 0.5 | margin |

In order to verify the effectiveness of the calculation method for the instantaneous posture of the milling cutter and the minor cutting edge of the cutter teeth, change the milling cutter speed but with the cutting efficiency unchanged, and carry out the milling experiment on titanium alloy.

In order to eliminate the influence of the three cutter tooth error changes caused by the replacement of cutter teeth on the friction and wear boundary of the minor flank and also to obtain the vibration data during the cutter tooth wear process, new cutter teeth were used for each experimental scheme during the experiment, and the milling vibration signal was measured in real time in the experiments. The experimental protocol is shown in Table 3.

**Table 3.** Milling parameters.

| Experiment Serial | $a_e$ (mm) | $a_p$ (mm) | $v_f$ (mm/min) | $n$ (r/min) |
|---|---|---|---|---|
| Scheme 1 | 16 | 0.5 | 275 | 687 |
| Scheme 2 | 16 | 0.5 | 275 | 713 |

Before the experiments, the axial and radial errors of the milling cutter teeth were measured by a tool setter, as shown in Table 4.

**Table 4.** Radial and axial errors of cutter teeth.

| Experiment | $\Delta r_i$ (mm) | | | $\Delta Z_i$ (mm) | | |
|---|---|---|---|---|---|---|
| | Cutter Tooth 1 | Cutter Tooth 2 | Cutter Tooth 3 | Cutter Tooth 1 | Cutter Tooth 2 | Cutter Tooth 3 |
| Scheme 1 | 0 | 0.012 | 0.029 | 0 | 0.024 | 0.012 |
| Scheme 2 | 0.017 | 0 | 0.003 | 0.007 | 0.013 | 0 |

In the experiments, the charge output PCB triaxial acceleration sensor was placed on the upper surface of the workpiece to collect real-time vibration acceleration signals. The vibration acceleration signal of the last feeding of scheme 1 and scheme 2 is shown in Figure 4.

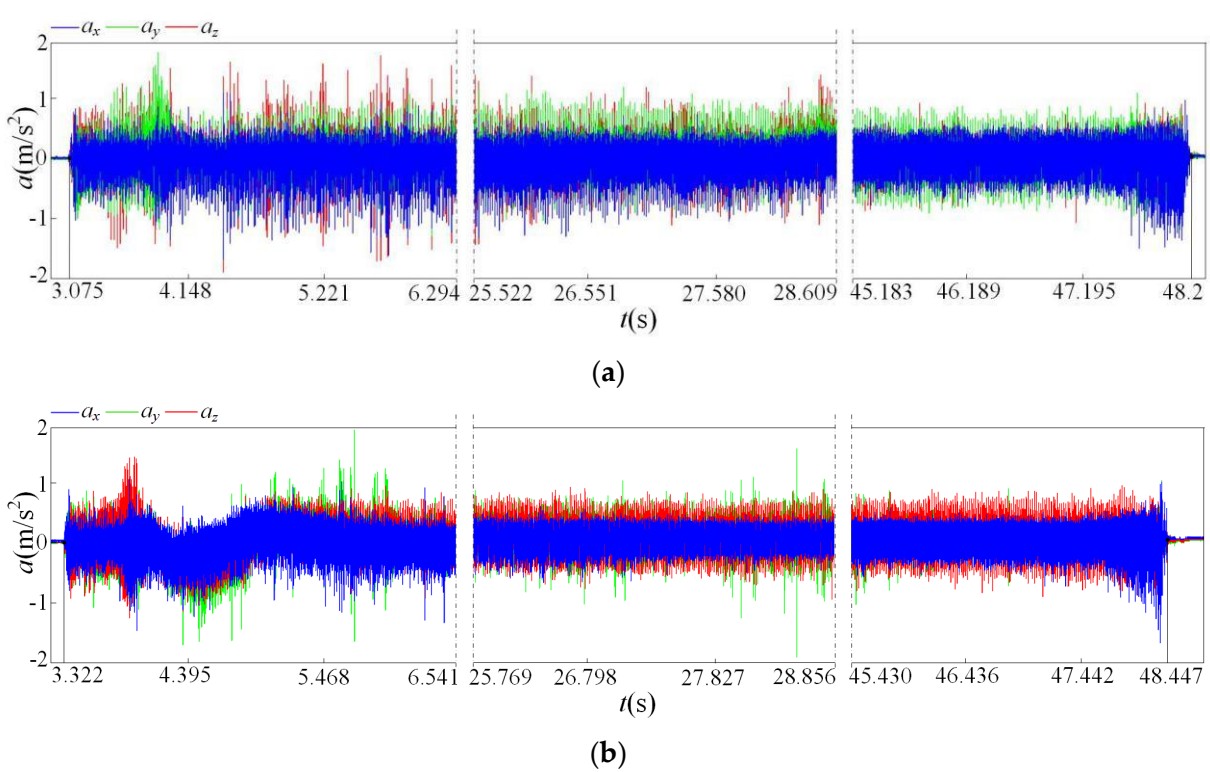

(**a**)

(**b**)

**Figure 4.** Milling vibration time domain signals. (**a**) Experimental scheme 1; (**b**) Experimental scheme 2.

After the milling experiments, the surface morphology was obtained by the white light interferometer, as shown in Figure 5a. Using Equations (1)–(11), the simulation result of the workpiece surface morphology was obtained, as shown in Figure 5b.

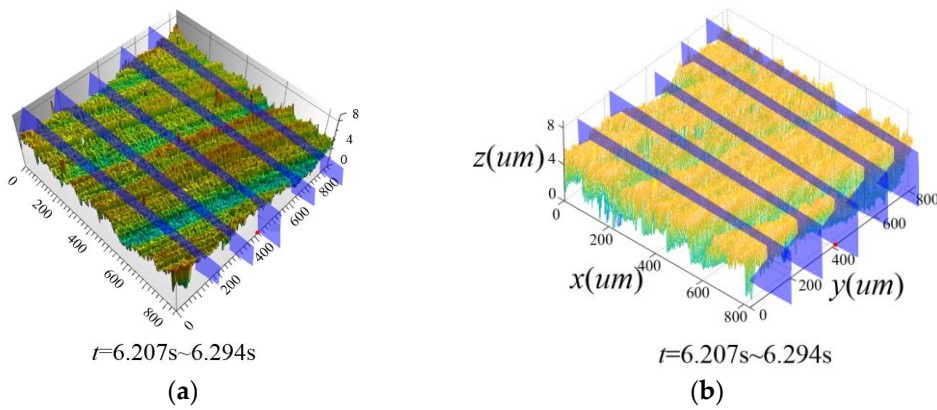

t=6.207s~6.294s

(**a**)

t=6.207s~6.294s

(**b**)

**Figure 5.** Comparison of experimental and simulation results of the machined surface morphology. (**a**) Experimental results; (**b**) Solution result.

Using the grey relative correlation analysis method [20], correlation analysis was carried out on the characteristic curves of the machined surface morphology on the corresponding sections in Figure 5a,b. The obtained correlation was greater than 0.80, which means the simulation results of the machined surface morphology were close to the experimental results, which proved the effectiveness of the milling and cutter tooth instantaneous posture calculation method under milling vibration.

## 3. Instantaneous Contact between the Minor Flank and Transition Surface

Based on the analysis results of the instantaneous posture solution model of the milling cutter and the minor cutting edge of the cutter teeth and the analysis results of the formation process of the workpiece surface morphology, the contact relationship between the cutter teeth and the workpiece was characterized, as shown in Figure 6.

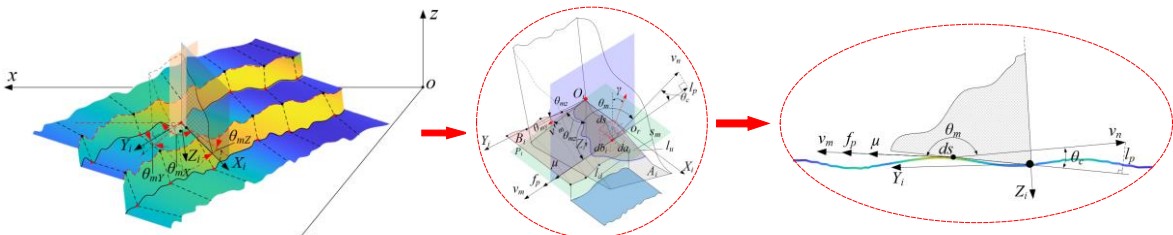

**Figure 6.** Instantaneous contact relationship between the minor flank and transition surface.

The equation of the workpiece machining transition surface in the workpiece coordinate system could be expressed as:

$$B_i(x,y,z,t) = \begin{cases} l'_f(x,y,z,t) = [x(t),y(t),z(t)] \\ t_i^s \leq t \leq t_i^e \end{cases} \tag{12}$$

$$l'_f(x,y,z,t) = \vartheta \cdot l_f(X_i,Y_i,Z_i) \tag{13}$$

$$A'_i(x,y,z,t) = \vartheta \cdot A_i(X_i,Y_i,Z_i) \tag{14}$$

where $A'_i$ is the minor flank of the cutter tooth in the workpiece coordinate system. $l'_f$ is the minor cutting edge in the workpiece coordinate system. $t_i^s$ is the moment when the tool cuts into the workpiece instantaneously. $t_i^e$ is the instant when the cutting tool cuts out the workpiece.

In the cutter tooth coordinate system, $d_s$ are the surface elements of the minor flank of the cutter tooth, as shown in Equation (15).

$$ds(X_i, Y_i, Z_i) = \frac{1}{\cos(\gamma(X_i, Y_i, Z_i))} \cdot dX_i \cdot dY_i \tag{15}$$

The angle between the normal vector and the Z axis in the coordinate system $O_i$-$X_iY_iZ_i$ was expressed as follows:

$$\gamma(X_i, Y_i, Z_i) = \arccos \frac{1}{\sqrt{1 + \left(\frac{\partial A_i(X_i,Y_i,Z_i)}{\partial X_i}\right)^2 + \left(\frac{\partial A_i(X_i,Y_i,Z_i)}{\partial Y_i}\right)^2}} \tag{16}$$

From Equations (12)–(16), because the vibration and cutter tooth errors change the instantaneous friction contact state of the cutter, this further leads to the changes in the instantaneous posture of the friction surface elements of the minor flank.

Using the square shoulder milling cutter, cutting parameters, tool tooth error and vibration detection data in the experimental scheme for milling titanium alloys and the calculation results of the milling cutter's instantaneous cutting behavior, the finite element model and boundary conditions of the square shoulder milling cutter for milling titanium alloys were constructed. The Johnson–Cook constitutive parameters and boundary conditions of the TC4 titanium alloy material are shown in Tables 5 and 6.

**Table 5.** Parameters of J–C constitutive model of TC4 titanium alloy.

| Material | *A* (Mpa) | *B* (Mpa) | *e* | *c* | *m* |
|----------|-----------|-----------|-----|-----|-----|
| TC4 | 860 | 683 | 0.47 | 0.035 | 1.0 |

**Table 6.** Boundary conditions of analysis scheme 1.

| Tool Material | Coating Material | Thermal Conductivity $\lambda$ (w/m·°C) | Rotating Speed *n* (r/min) | Feed Rate | Cutting Width $a_e$ (mm) | Cutting Depth $a_p$ (mm) |
|---------------|------------------|-----------------------------------------|----------------------------|-----------|-------------------------|--------------------------|
| WC | Ti-CN | 45 | 687 | 275 | 16 | 0.5 |

In Table 5, *A*, *B*, *e*, *c* and m are the yield stress strength, strain hardening constant, strain hardening exponent, strain rate hardening parameter and temperature strain rate sensitivity, respectively.

The analysis results for the instantaneous thermal-stress coupling field on the flank of the cutter tooth in the Deform-3D environment, as shown in Figure 7.

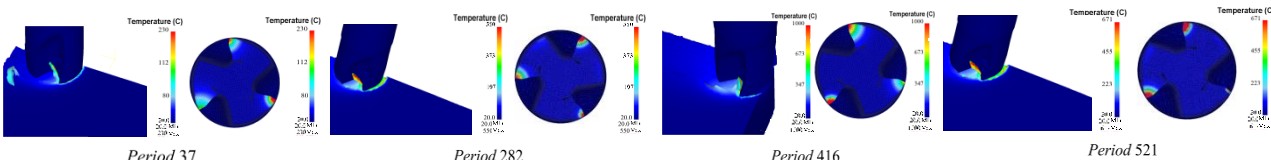

*Period* 37     *Period* 282     *Period* 416     *Period* 521

**Figure 7.** Instantaneous thermal-stress coupling field of cutter tooth 1 of scheme 1 ($\varphi_i = 52°$).

According to the analysis results of material mechanics, if the equivalent stress on the minor cutting edge of the cutter tooth ($\sigma$) greater than the yield strength of the material ($\sigma_S$), the material in the cutting edge area would fall off. Therefore, the equivalent stress could be used as the upper boundary criterion for the instantaneous friction of the cutter teeth, as shown in Equation (17).

$$\sigma \geq \sigma_s \tag{17}$$

In the whole milling process, there was equivalent strain in the contact area and the non-contact area of the minor flank of the cutter tooth. The equivalent strain in the friction contact area decreases gradually along the direction of the minor flank of the cutter tooth and changes abruptly at the lower boundary of the instantaneous friction. Therefore, the lower boundary node of the friction of the minor flank of the cutter tooth was identified by the equal effect change rate, as shown in Equation (18).

$$\varepsilon' = \frac{d\varepsilon(Y_i(t))}{dY_i(t)}, \varepsilon' \geq \varepsilon_s' \tag{18}$$

where $\varepsilon$ is the equivalence strain. $\varepsilon'$ is the equivalent strain rate. $\varepsilon_s'$ is the threshold of the equivalent strain rate mutation.

Therefore, using Equations (17) and (18), the upper and lower boundaries of the instantaneous friction on the flank could be obtained, as shown in Equation (19).

$$\begin{cases} l_u(x(t), y(t), z(t)) = 0 \\ l_d(x(t), y(t), z(t)) = 0 \end{cases} \tag{19}$$

The identification method for the cumulative friction boundary at the minor flank section of the cutter tooth was as follows:

$$Y_{i\max} = \max(Y_i(t)), G(X_i, Y_{i\max}) = 0 \tag{20}$$

where $Y_i$ is the instantaneous boundary at different positions of $X_i$. $Y_{i\max}$ is the maximum of $Y_i$.

Using Equations (17)–(20), the calculation results for the cumulative friction boundary of the three cutter teeth in the analysis scheme 1 were obtained. Using the ultra-depth-of-field microscope, the experimental results for the wear boundary of the minor flank of the cutter tooth were obtained. Figure 8 shows the friction and wear detection results for the flank friction boundary of the cutter tooth in the analysis scheme 1 and the friction and wear test results for the experimental scheme 1.

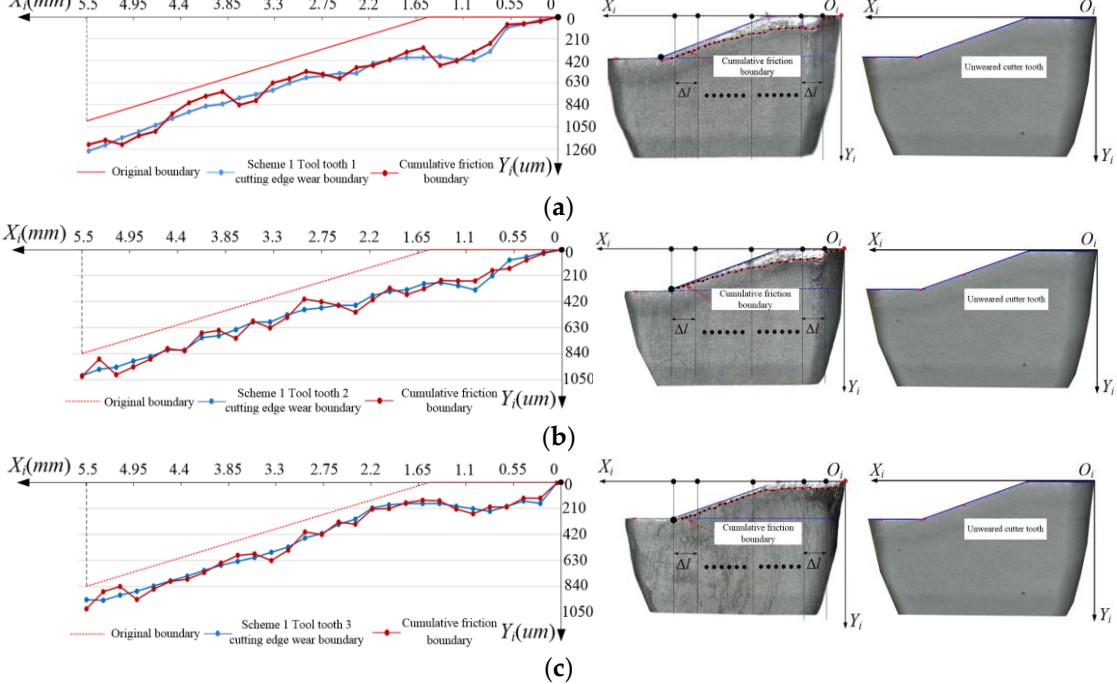

**Figure 8.** Comparison between the simulated and experimental wear accumulation boundary of the minor flank of scheme 1. (**a**) Cutter tooth 1; (**b**) Cutter tooth 2; (**c**) Cutter tooth 3.

In order to further verify the correctness of the calculated cumulative friction boundaries, the correlation coefficients between the calculated and measured cumulative friction boundaries was obtained, as shown in Equations (21)–(24):

$$\rho = \frac{Cov(X_i, Y_i)}{\sigma_{X_i} \cdot \sigma_{Y_i}} \tag{21}$$

$$Cov(X_i, Y_i) = E\left[(X_i - \mu_{x_i})(Y_i - \mu_{y_i})\right] \tag{22}$$

$$\sigma_{X_i} = \sqrt{E((X_i - \mu_{x_i})^2)} \tag{23}$$

$$\sigma_{Y_i} = \sqrt{E((X_i - \mu_{y_i})^2)} \tag{24}$$

where $X_i$ and $Y_i$ are the measured values of each characteristic point of the boundary curve in the cutter tooth measurement coordinate system. $\mu_{x_i}$ and $\mu_{y_i}$ are the average values of the boundary curve of each point.

From Equations (21)–(24), the correlation coefficients of the two kinds of boundaries of the three cutter teeth in scheme 1 were 0.82, 0.87 and 0.90, respectively, which proved the accuracy of the model of the cumulative friction boundary. The above friction boundary solution method can be used to reveal the dynamic characteristics of the flank friction of the cutter tooth.

## 4. Variation Characteristic of Instantaneous Friction Velocity Speed on the Minor Flank of the Cutter Tooth

In order to reveal the change characteristics of the instantaneous friction speed vector on the minor flank of cutter tooth, according to the instantaneous friction upper and lower boundary criteria on the minor flank of the cutter tooth, the instantaneous contact friction area element between the minor flank of cutter tooth and the machining transition surface was identified. The friction speed size and direction angle of the surface element were solved.

The trajectory parameter equation of any point on the cutter tooth was obtained by using Equation (6). The trajectory of any point on the cutter tooth is skewed to time, and the sub-velocities $v_{nx}$, $v_{ny}$ and $v_{nz}$ along the $x$, $y$ and $z$ axes of the workpiece coordinate were obtained, respectively.

The relative speed $v_n$ of any point is:

$$v_n = \sqrt{v_{nx}^2 + v_{ny}^2 + v_{nz}^2} \tag{25}$$

The unit vector of the relative speed $v_n$ in the workpiece coordinate is $(v_{nx}, v_{ny}, v_{nz})$. The intersection point between the minor flank of the cutter tooth and the workpiece transition surface is $o_r$. The common tangent plane $P_i$ passes through the or point and is tangent to the minor flank of the cutter tooth and the workpiece transition surface at the same time:

$$P_i(x, y, z) = \frac{\partial A_i'(x, y, z)}{\partial x}(x - x_r) + \frac{\partial A_i'(x, y, z)}{\partial y}(y - y_r) + \frac{\partial A_i'(x, y, z)}{\partial z}(z - z_r) \tag{26}$$

Project $v_n$ on the common tangent plane $P_i$, and the unit vector $\overrightarrow{l}_p$ in the projection direction through the point or is $(l_{px}, l_{py}, l_{pz})$.

In the workpiece coordinate, the friction speed $\boldsymbol{v_m}$ at any point on the minor flank is as follows:

$$v_m = v_n \cdot \cos\theta_m \tag{27}$$

In the equation, $\theta_m$ is the angle between the relative velocity $v_n$ and the unit vector $\vec{l}_p$. $\theta_m$ and $\theta_c$ are complementary angles to each other. The calculation method of $\theta_c$ is as follows:

$$\theta_c = \arccos(\frac{l_{px} \cdot v_{nx} + l_{py} \cdot v_{ny} + l_{pz} \cdot v_{nz}}{\sqrt{l_{px}^2 + l_{py}^2 + l_{pz}^2} \cdot \sqrt{v_{nx}^2 + v_{ny}^2 + v_{nz}^2}}) \tag{28}$$

In Equation (28), $\theta_c$ is the angle between the relative speed $v_n$ and the friction speed $v_m$.

In the workpiece coordinate, the friction speed direction vector $\vec{v}_m$ of the surface element on the minor flank is $(v_{mx}, v_{my}, v_{mz})$. The unit vectors of the $y$-axis and $z$-axis of the workpiece coordinate are $\vec{l}_{ry}$ and $\vec{l}_{kz}$, respectively. The angle $\theta_{my}$ and $\theta_{mz}$ between $v_m$ and the $y$-axis and $z$-axis of the cutter tooth coordinate system are obtained by the two-vector angle solution method, respectively.

According to the instantaneous transformation relationship matrix between the cutter tooth coordinate and the workpiece coordinate, the friction speed components $v_{mXi}$, $v_{mYi}$ and $v_{mZi}$ of the instantaneous friction surface element of the minor flank of the cutter tooth were obtained, as shown in Equation (29).

$$\begin{bmatrix} v_{mXi} & v_{mYi} & v_{mZi} & 1 \end{bmatrix}^T = \vartheta^{-1} \cdot \begin{bmatrix} v_{mx} & v_{my} & v_{mz} & 1 \end{bmatrix}^T \tag{29}$$

In the equation, $v_{mXi}$ is the friction speed of the surface element on the minor flank of the cutter tooth along the $X_i$ axis of the cutter tooth coordinate. $v_{mYi}$ is the friction speed of the surface element on the minor flank of the cutter tooth along the $Y_i$ axis of the cutter tooth coordinate. $v_{mZi}$ is the friction speed of the surface element on the minor flank of the cutter tooth along the $Z_i$ axis of the cutter tooth coordinate.

The friction speed at any point in the cutter tooth coordinate is as follows:

$$v_m(X_i, Y_i, Z_i) = \sqrt{v^2_{mXi} + v^2_{mYi} + v^2_{mZi}} \tag{30}$$

Using the unit vectors of the cutter tooth coordinate for the $Y_i$ axis and $Z_i$ axis of $\vec{l}_r$ and $\vec{l}_k$, the $\theta_{mYi}$ and $\theta_{mZi}$ of the angle between $v_m$ and the tool tooth coordinate $Y_i$ axis and $Z_i$ axis were obtained, as shown in Equations (31) and (32).

$$\theta_{mYi} = \arccos(\frac{l_{rYi} \cdot v_{mYi}}{|l_{rYi}| \cdot \sqrt{v_{mXi}^2 + v_{mYi}^2 + v_{mZi}^2}}) \tag{31}$$

$$\theta_{mZi} = \arccos(\frac{l_{kZi} \cdot v_{mZi}}{|l_{kZi}| \cdot \sqrt{v_{mXi}^2 + v_{mYi}^2 + v_{mZi}^2}}) \tag{32}$$

Using Equations (25)–(32), the friction velocity vector declination angles and their magnitudes for the three tooth minor flank area units $ds(0.3, 0.05, 0)$ in experimental scheme 1 were calculated, and the results are shown in Figure 9.

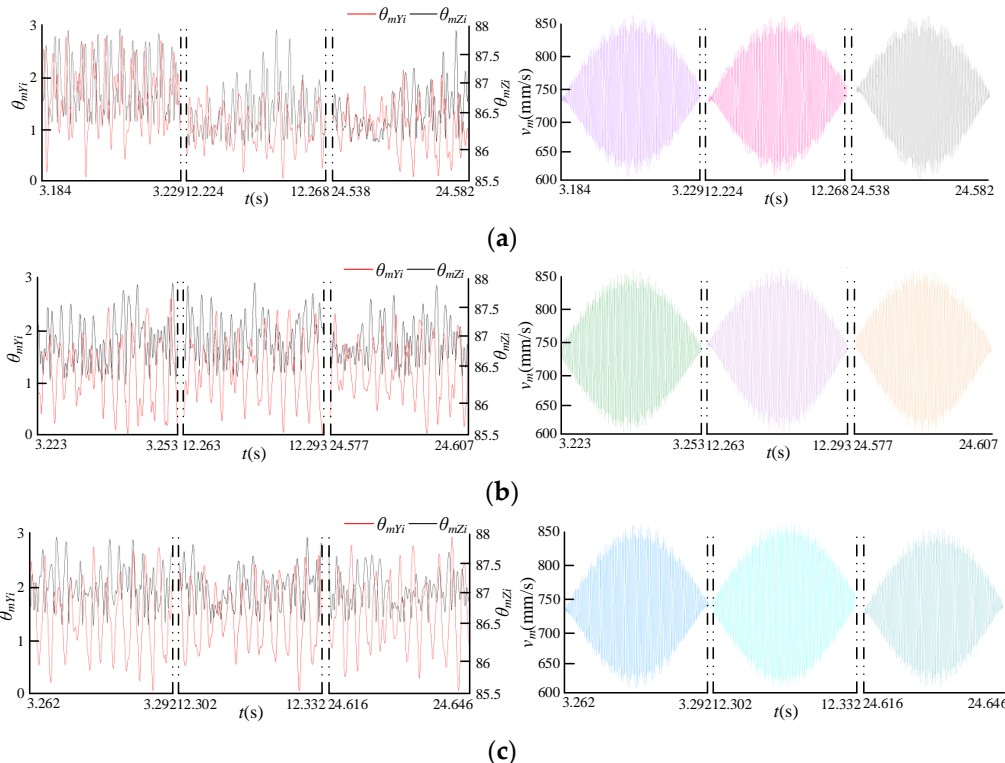

**Figure 9.** Variation in the friction velocity vector of the tooth minor flank area unit ds(0.3, 0.05, 0). (**a**) Cutter tooth 1; (**b**) Cutter tooth 2; (**c**) Cutter tooth 3.

It could be seen from Figure 9 that under the milling vibration and cutter tooth error, the overall change trend of the friction speed of each cutter tooth was the same, and it gradually increases to the maximum value and then gradually decreases. However, there were obvious differences in the variation characteristics of the friction speed vector declination angle of each cutter tooth. This directly affects the variation characteristics of the friction speed of each tooth.

According to the analysis results of the thermal-stress coupling field of the square shoulder milling cutter milling titanium alloy, Equations (25)–(30) were used to calculate the friction speed of the minor flank of the cutter tooth point by point during the cutting process, and the minor flank friction of the cutter tooth was obtained. The velocity distribution results are shown in Figure 10.

From Figures 4, 9 and 10, in different milling periods determined by the rotating speed of the milling cutter, although the instantaneous position angle of the same cutter tooth was the same, the friction velocity distribution of the minor flank of the cutter tooth was obviously different. The main reason was tha, the milling vibrations at different milling moments corresponding to the same instantaneous position angle of the cutter teeth were different, which makes the instantaneous posture of the cutter tooth minor flank significantly different.

It can be seen from the comparison between Figure 10 and Table 4, in the same milling period and at the same instantaneous position angle, the friction velocity distribution of the flank surface of different teeth was significantly different. The main reason was that, due to the influence of the cutter tooth error and the angle between the cutter teeth, the instantaneous contact between the minor flank of each cutter tooth and the machining transition surface has axial and radial offsets. Eacg cutter tooth also corresponds to the same instantaneous position angle with the milling vibration of different milling moments.

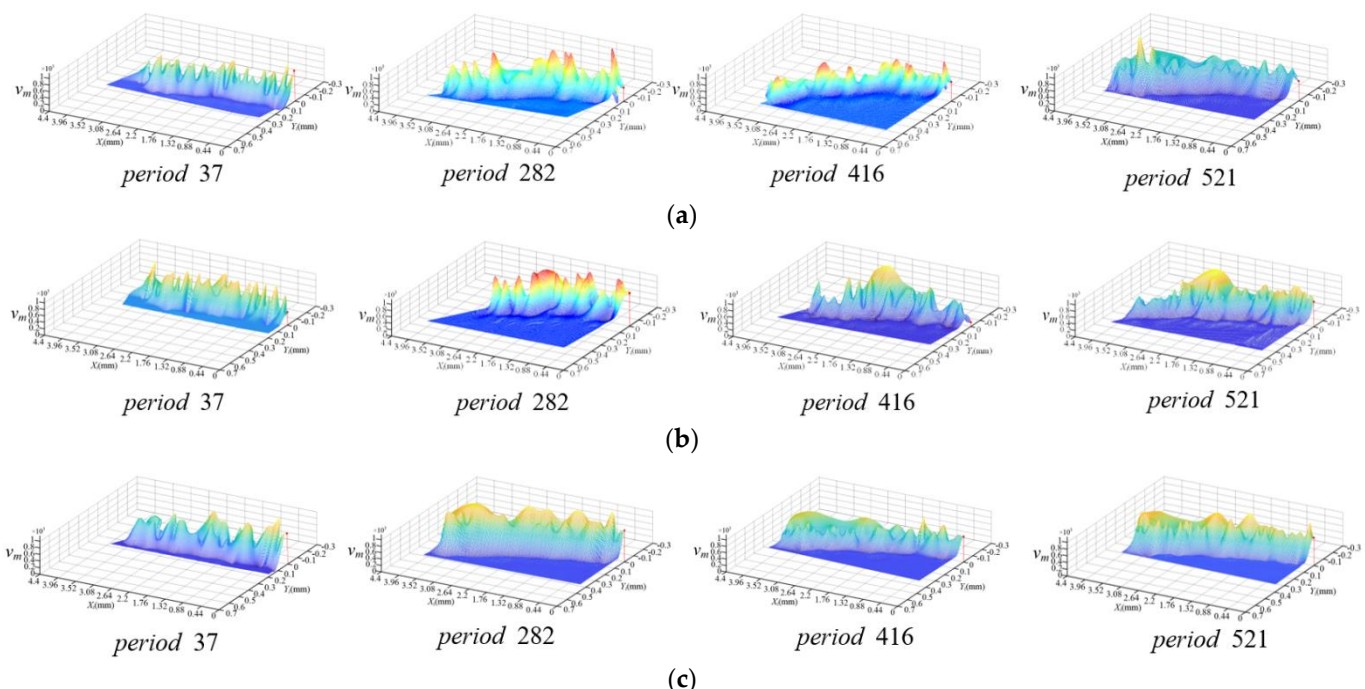

**Figure 10.** The instantaneous friction speed distribution on the minor flank in scheme 1 ($\varphi_i = 52°$). (**a**) Cutter tooth 1; (**b**) Cutter tooth 2; (**c**) Cutter tooth 3.

The above analysis results showed that the instantaneous friction velocity vector of the minor flank would be obviously affected by cutter tooth error and the milling vibration, and the variations of the instantaneous friction velocity vector were different. The result will lead to the variability of the friction process of the minor flank of each tooth of the milling cutter.

## 5. Instantaneous Friction Energy Consumption of the Minor Flank of the Cutter Tooth

The instantaneous contact between the tool and the workpiece produced a large amount of energy consumption, and energy consumption was one of the main factors leading to tool wear. In order to research the dynamic characteristics of instantaneous friction energy consumption on the minor flank of cutter tooth, the energy conversion relationship between the interface atoms of tool and workpiece was used to construct the model of friction energy consumption, as shown in Formulas (33)–(36).

According to interface atomic theory, the $E_v$, which is the change rate of absorbed energy, is as follows:

$$E_v(a_i, b_i, c_i) = \frac{dE}{dt} = \frac{2ds(a_i, b_i, c_i) \cdot v_m(a_i, b_i, c_i, t) \cdot h \cdot v}{0.5a^3 \cdot \left(e^{\frac{h \cdot v}{k \cdot T(a_i, b_i, c_i, t)}} - 1\right)} \tag{33}$$

In Formula (33), $E$ represents the energy absorbed at time t, and $v$ represents the frequency of atomic forced vibration, as shown in Formula (34).

$$v = \frac{v_m(a_i, b_i, c_i, t)}{a} \tag{34}$$

The absorbed instantaneous energy distribution function is shown in Formula (35).

$$E_s(a_i, b_i, c_i, t) = \int_{t_{es1}}^{t_{es2}} \frac{2ds(a_i, b_i, c_i) \cdot v_m(a_i, b_i, c_i, t) \cdot h \cdot v}{0.5a^3 \cdot \left(e^{\frac{h \cdot v}{k \cdot T(a_i, b_i, c_i, t)}} - 1\right)} dt \tag{35}$$

Among them: *a* is the lattice constant, which is $2.9506 \times 10^{-10}$ m, *h* is the Planck constant, which is $6.62607015 \times 10^{-34}$ J·s, *k* is the Boltzmann constant, which is $1.380649 \times 10^{-23}$ J/K) and tes1 and tes2 are the instantaneous initial time and termination time of the absorbed energy, respectively. *T* is the temperature rise of atomic interface. The calculation method is as follows in Formula (36):

$$T(a_i, b_i, c_i, t) = \frac{2}{m \cdot k} \left( \frac{\pi \cdot a \cdot v_m(a_i, b_i, c_i, t) \cdot H}{a^2 \cdot \omega_n^2 - 4\pi^2 \cdot v_m^2(a_i, b_i, c_i, t)} \right) \tag{36}$$

Among them: m is the relative atomic mass of the atom, which is $4.34 \times 10^{-26}$ kg, $\omega_n$ is the natural frequency of the atom, which is $4.39 \times 10^{11}$ rad/s, and H is the excitation force of the interface potential energy field, which is $1 \times 10^{-9}$ N.

As shown in Equations (33)–(36), the friction energy distribution of the minor flank of the cutter tooth is obtained, as shown in Figure 11.

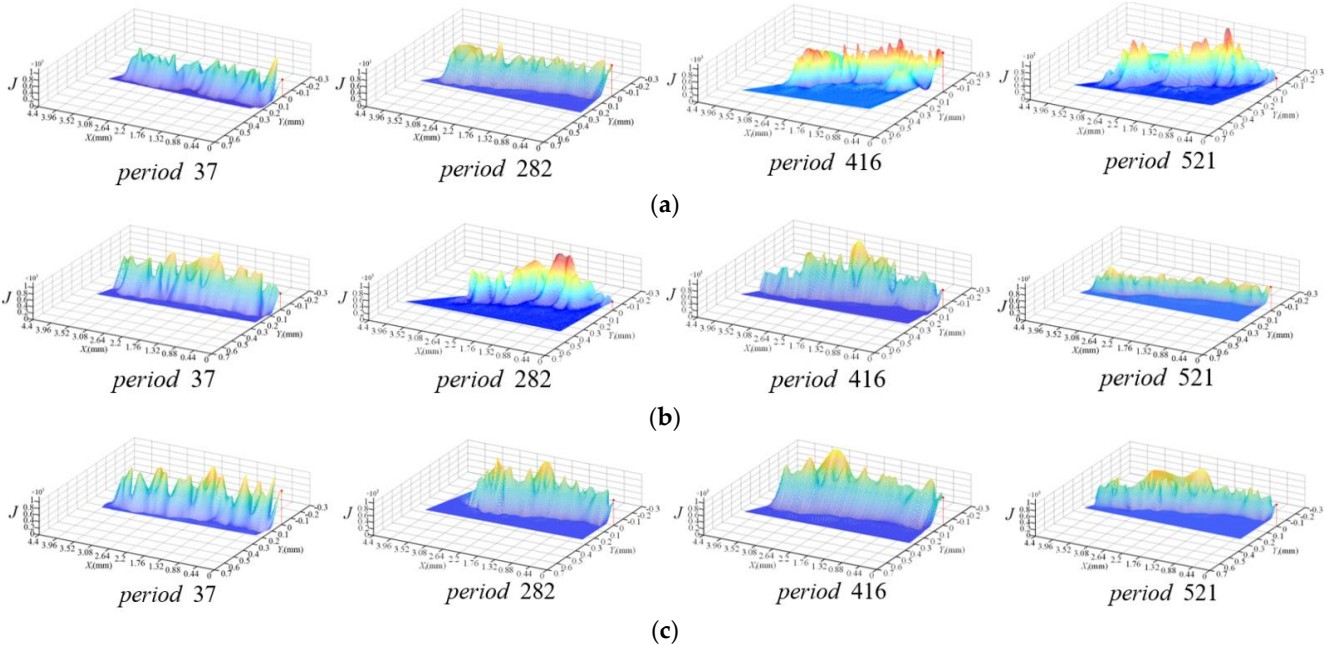

**Figure 11.** Distribution of instantaneous frictional energy consumption on the minor flank of the tooth of scheme ($\varphi_i = 52°$). (**a**) Cutter tooth 1; (**b**) Cutter tooth 2; (**c**) Cutter tooth 3.

It can be seen from Figures 7 and 11 that, with the changes in the cutter tooth error and the milling vibration, the instantaneous friction velocity vector, instantaneous contact stress and contact temperature of the minor flank of the cutter tooth constantly changed, which leads to the changes in the friction energy distribution boundary. The energy consumption showed significantly different variations. This means that not only the dynamics of the instantaneous friction change significantly, but also the difference in the friction process of the minor flank of each cutter will change. It is necessary to further identify its friction characteristic variables.

## 6. Instantaneous Friction Coefficient and Friction Stress Distribution of the Minor Flank of the Cutter Tooth

According to Figures 2 and 3, in the space defined by the cutter tooth coordinate system, the relationship between the instantaneous equivalent stress and the instantaneous normal stress is shown in Figure 12.

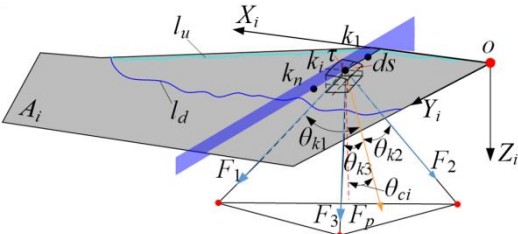

**Figure 12.** Equivalent stress decomposition.

In Figure 12, $Fp$ is the normal stress on the minor flank area unit of the cutter tooth. The angle between the vertical node and the minor flank of the cutter tooth and the $c_i$ axis is $\theta_{ci}$. $\tau$ is the equivalent stress of the minor flank of the cutter tooth at the network node under the thermal coupling field. $F_1$, $F_2$ and $F_3$ are the equivalent stresses in the direction of the tetrahedral area element, respectively.

The equivalent stress and normal stress are represented by vectors in the cutter tooth coordinate system, as shown in Equations (37) and (38).

$$\begin{cases} \vec{F}_1 = (a_{F1}, b_{F1}, c_{F1}) \\ \vec{F}_2 = (a_{F2}, b_{F2}, c_{F2}) \\ \vec{F}_3 = (a_{F3}, b_{F3}, c_{F3}) \end{cases} \tag{37}$$

$$\vec{F}_p = \left(0, 0, \sqrt{a_i^2 + b_i^2 + c_i^2}\right) \tag{38}$$

From Equations (37) and (38), $\theta_{k1}$, $\theta_{k2}$, and $\theta_{k3}$ can be expressed as:

$$\begin{cases} \theta_{k1} = \arccos(\frac{\vec{F}_p}{\vec{F}_1}) \\ \theta_{k2} = \arccos(\frac{\vec{F}_p}{\vec{F}_2}) \\ \theta_{k3} = \arccos(\frac{\vec{F}_p}{\vec{F}_3}) \end{cases} \tag{39}$$

According to Equations (37)–(39), the normal stress was obtained as:

$$F_p(a_i, b_i, c_i) = F_1(a_i, b_i, c_i) \cdot \cos\theta_{k1} + F_2(a_i, b_i, c_i) \cdot \cos\theta_{k2} + F_3(a_i, b_i, c_i) \cdot \cos\theta_{k3} \tag{40}$$

$F_n$ is the normal pressure of the friction contact area unit, and $\mu$ is the friction coefficient. Then, the work performed by the unit friction force in the $dt$ can be expressed as:

$$dW = \mu(a_i, b_i, c_i, t) \cdot F_n(a_i, b_i, c_i, t) \cdot v_m(a_i, b_i, c_i, t) \cdot dt \tag{41}$$

It was assumed that during the frictional motion of the tool–worker interface, all the frictional work was converted into the thermal energy of the system.

$$dW = dE \tag{42}$$

From Equations (37)–(42), the friction coefficient distribution function is shown in Equation (43).

$$\mu(a_i, b_i, c_i, t) = \frac{2h \cdot v(a_i, b_i, c_i, t)}{0.5a^3 \cdot F_p(a_i, b_i, c_i, t) \cdot \left(e^{\frac{h \cdot v}{k \cdot T(a_i, b_i, c_i, t)}} - 1\right)} \tag{43}$$

The friction stress distribution function obtained from Equations (41)–(43) is shown in Equation (44).

$$f_p(a_i, b_i, c_i, t) = \mu(a_i, b_i, c_i, t) \cdot F_p(a_i, b_i, c_i, t) \tag{44}$$

According to Equation (38), the friction coefficient distribution of the minor flank of the cutter tooth was calculated, as shown in Figure 12.

From the above analysis, the friction coefficient was mainly affected by the friction speed and the normal stress. Comparing Figures 8 and 13, the friction speed and normal stress in the non-contact area between the minor flank of the cutter tooth and the machining transition surface of the workpiece was 0. So, the friction coefficient boundary was the same as the friction speed boundary. However, due to the different normal stress values in the frictional contact area, the instantaneous distribution of the friction coefficient on the minor flank and the distribution of the friction speed were different.

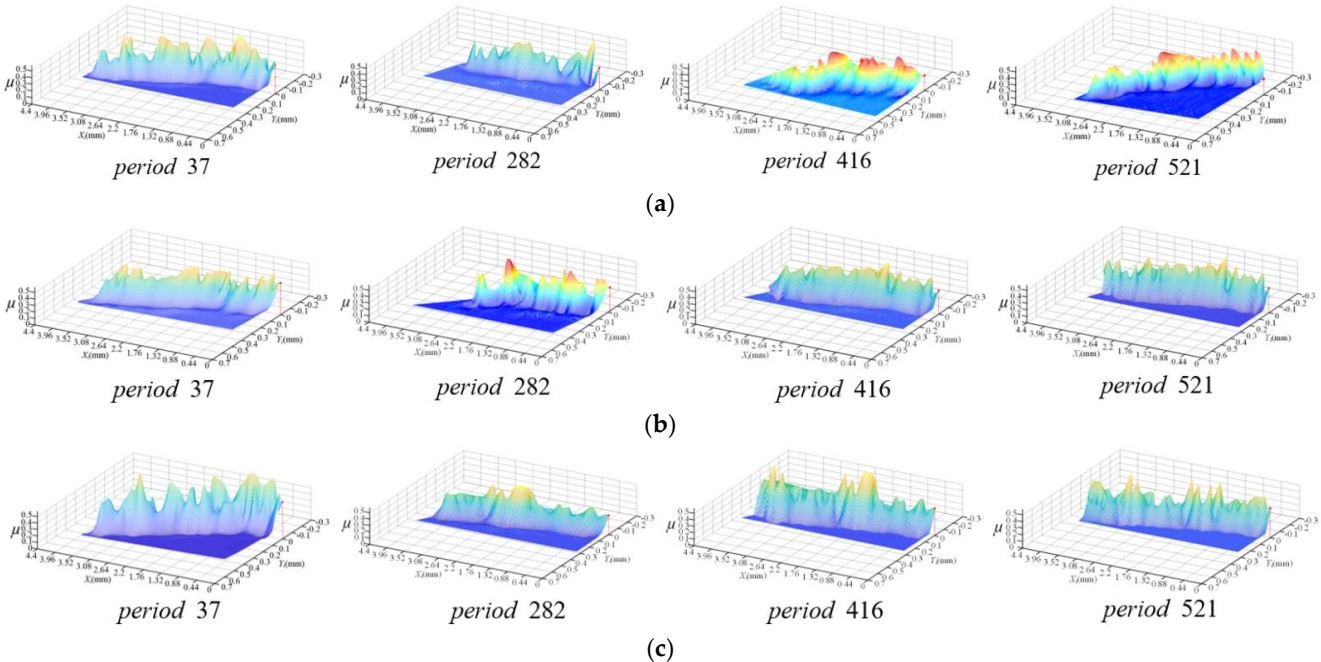

**Figure 13.** Instantaneous friction coefficient distribution on the minor flank of the cutter tooth of scheme 1 ($\varphi_i = 52°$). (**a**) Cutter tooth 1; (**b**) Cutter tooth 2; (**c**) Cutter tooth 3.

Comparing the distribution of the friction coefficient of the minor flank of the three teeth, it can be seen that the distribution area of the friction coefficient of the minor flank of each tooth was completely different from the boundary shape. It showed that it was greatly affected by the cutter tooth error and milling vibration.

According to Equation (44), the friction force distribution on the minor flank of the first cutter tooth was calculated, as shown in Figure 14.

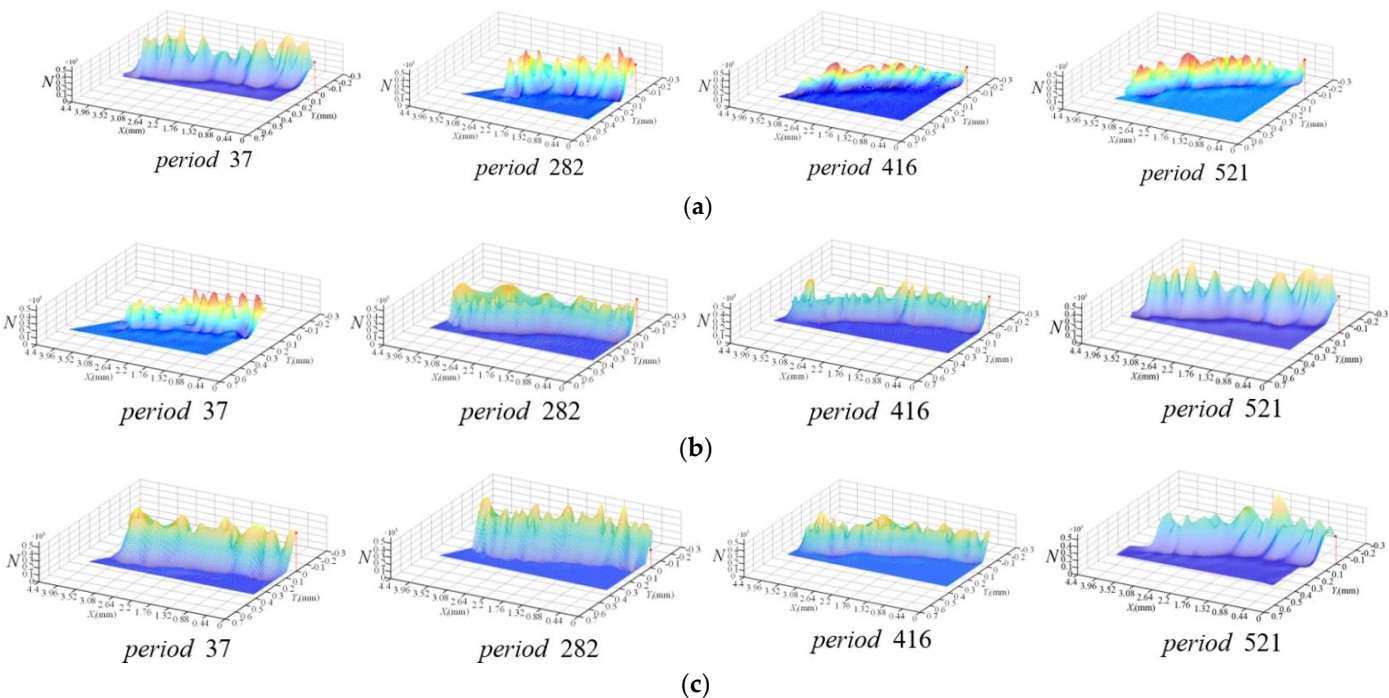

**Figure 14.** Instantaneous friction force distribution on the minor flank of the cutter tooth of scheme 1 ($\varphi_i = 52°$). (**a**) Cutter tooth 1; (**b**) Cutter tooth 2; (**c**) Cutter tooth 3.

The above analysis results showed that the variability in the flank friction process of the cutter tooth is not only reflected in the changes in the instantaneous contact, friction speed, and friction energy consumption, but also in the changes in the instantaneous coefficient of friction and the dynamics of friction force. Therefore, in order to reveal the friction process of the cutter tooth minor flank, it is necessary to comprehensively consider the relationship between the characteristic variables of the flank friction of the cutter tooth and its dynamic behavior, and identify the dynamic behavior of the characteristic variable of the flank friction of the cutter tooth with the action of vibration.

## 7. Dynamic Behavior Identification and Experimental Verification of Friction Characteristics Variables of Minor Flank Friction on the Cutter Tooth

According to the above analysis results, an identification method for the vibration of the friction behavior variables of the minor flank friction of the cutter tooth was proposed, as shown in Figure 15.

This method could unveil the dynamic characteristic of the friction variable of flack face of the cutter tooth based on the instantaneous posture of the flank area unit of the tooth and the instantaneous variables of the friction process. Using this method, the influence of key process variables on the friction velocity, energy consumption, friction coefficient and friction force could be identified. At the same time, since the method includes the structural parameters of the milling cutter and the cutter tooth error, the method is also suitable for the same type of high-efficiency milling cutter.

Using the milling cutter, cutting parameters and test data of experimental scheme 2 as analysis scheme 2, and using Equations (17)–(20), the friction and wear of experimental scheme 2 were obtained, as shown in Figure 16.

The correlation coefficients of the three cutter teeth in scheme 2 were 0.84, 0.88 and 0.86, respectively. This indicated that the friction energy consumption of the minor flank of the cutter tooth was highly correlated with the friction boundary of the cutter tooth measured by the experiment. It also verified the correctness of the friction energy consumption calculation method.

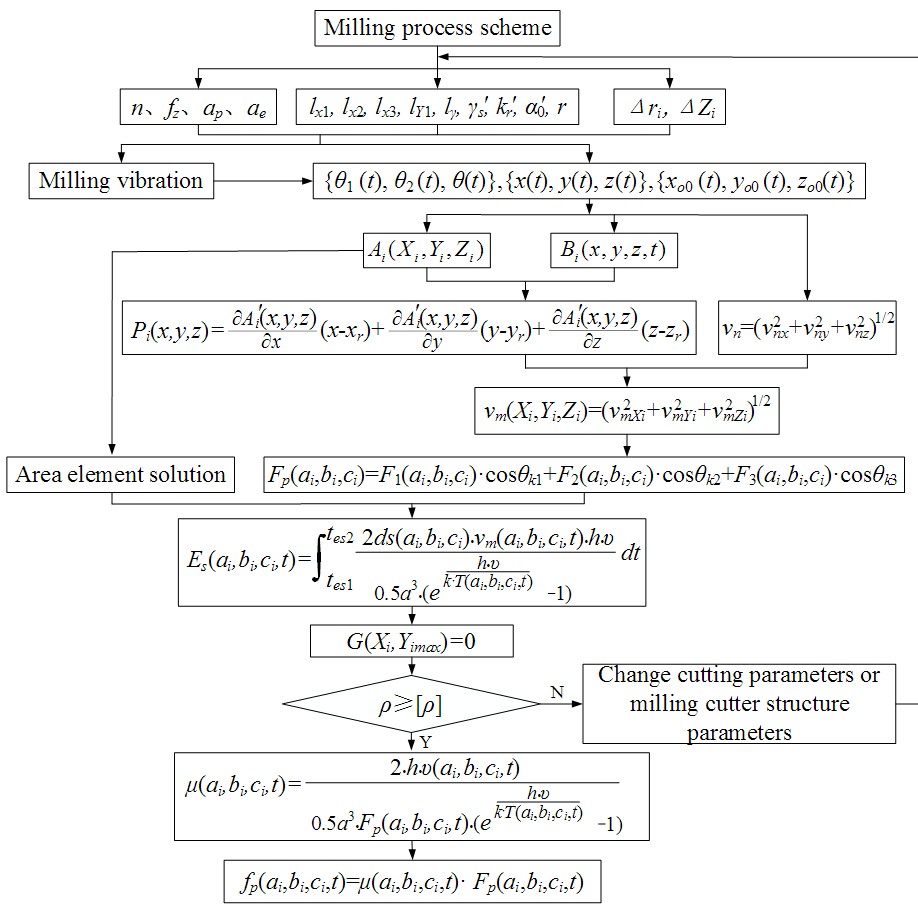

**Figure 15.** Identification of the variation in the friction behavior variables of the cutter tooth.

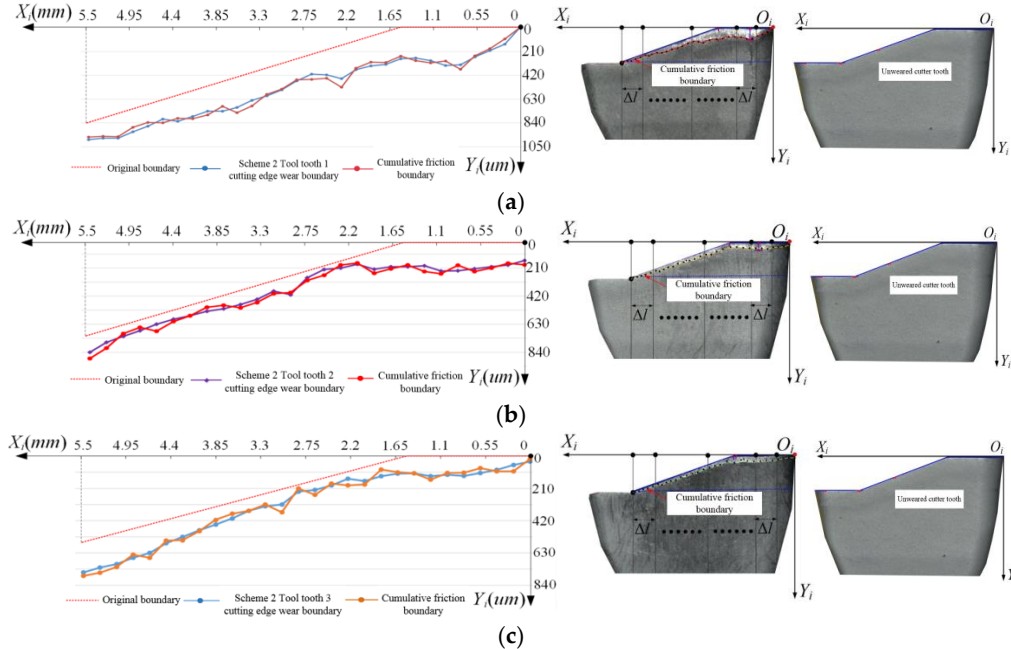

**Figure 16.** Comparison between the calculation and experimental friction accumulation boundary on the minor flank of the cutter tooth of scheme 2. (**a**) Cutter tooth 1; (**b**) Cutter tooth 2; (**c**) Cutter tooth 3.

According to the proposed model of the friction energy consumption and friction coefficient in Section 4, their distribution for three cutter teeth in scheme 2 are obtained, as shown in Figure 17.

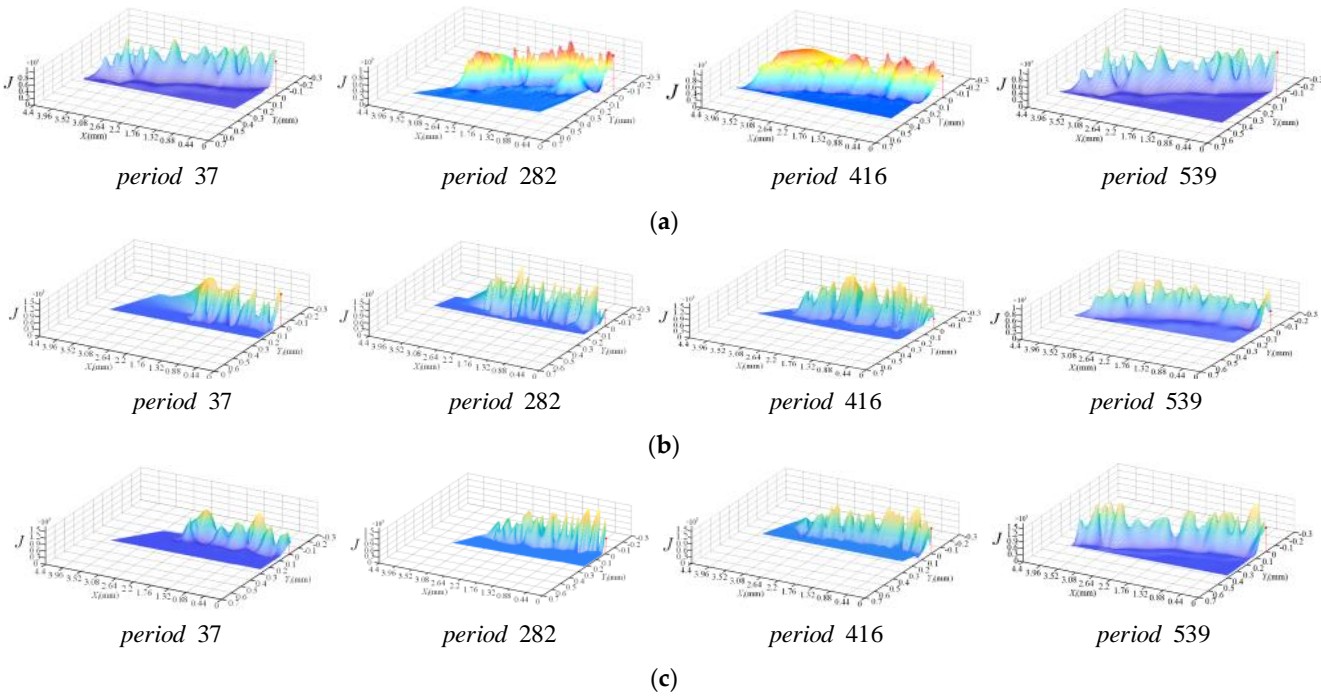

**Figure 17.** Instantaneous friction energy consumption distribution of the minor flank of cutter tooth for scheme 2 ($\varphi_i = 52°$). (**a**) Cutter tooth 1; (**b**) Cutter tooth 2; (**c**) Cutter tooth 3.

Using the friction coefficient calculation method, the distribution of the friction coefficient on the minor flank of the three cutter tooth was obtained, as shown in Figure 18.

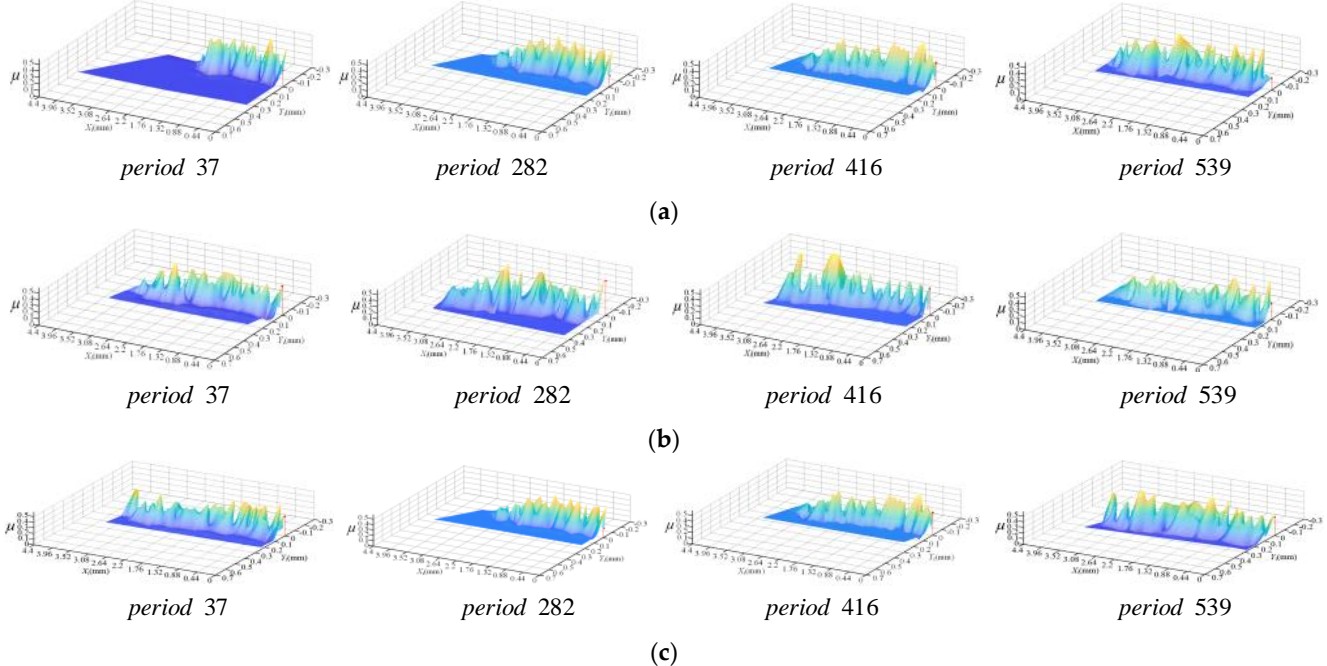

**Figure 18.** Instantaneous friction coefficient distribution of the minor flank of the cutter tooth for scheme 2 ($\varphi_i = 52°$). (**a**) Cutter tooth 1; (**b**) Cutter tooth 2; (**c**) Cutter tooth 3.

The kurtosis of the friction characteristic variable of the same feature point on the minor flank of the cutter tooth with time was calculated, and the kurtosis values of scheme 1 and scheme 2 were compared. The solution result for the feature point in Figure 9 is shown in Figure 19.

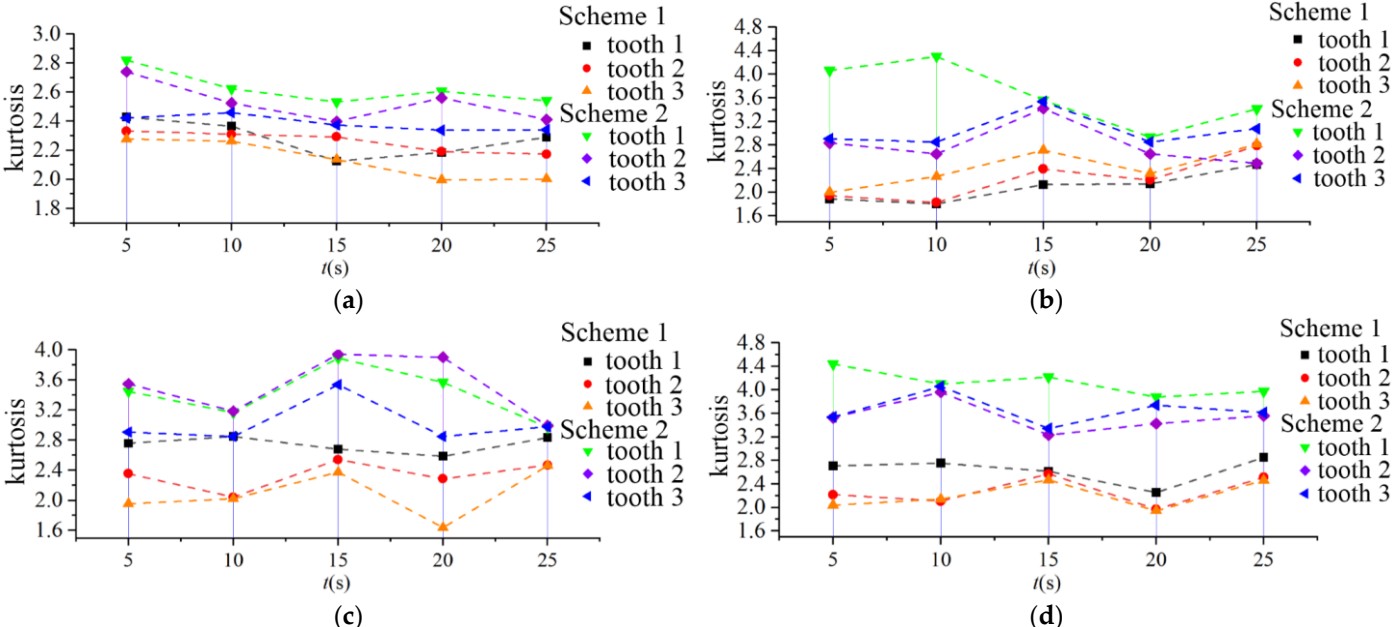

**Figure 19.** Variation kurtosis of the friction characteristic varibles in scheme 1 and scheme 2. (**a**) Friction speed; (**b**) Friction energy consumption; (**c**) Friction coefficient; (**d**) Friction force.

In Figure 19, the characteristic time of the time axis is the intermediate time corresponding to the sampling time during the last feed cutting of the milling cutter.

It can be seen from Figure 19 that the kurtosis of each parameter in scheme 2 is significantly greater than that in scheme 1 due to changes in the cutting parameters. This indicated the friction speed, friction energy consumption, friction coefficient, and friction force is sensitive to the change in cutting conditions. When the speed of scheme 2 increases, the curve variation range of the friction speed, friction energy consumption, friction coefficient and friction force was more obvious.

In summary, the models and methods established in this study could effectively identify the responses of the surface element friction variables of the minor flank of the cutter tooth to the process characteristic variables. It also provided a theoretical basis for the effective control of the wear of the milling cutter.

## 8. Conclusions

(1)  The instantaneous posture model of the friction area element of the minor flank of the cutter tooth was developed. The results showed that, due to the influence of the milling vibration and cutter tooth errors, the cutter tooth had a constantly changing displacement increment in three directions of the workpiece coordinate system. Therefore, the instantaneous cutting posture of the milling cutter and the cutter tooth surface element of the minor flank of the cutter tooth changed, which caused the instantaneous cutter-workpiece engagement to be in an unstable state.

(2)  A method for identifying the upper and lower boundaries of the instantaneous friction on the minor flank of the cutter tooth was proposed. The instantaneous contact relationship between the minor flank of the cutter tooth and the transition surface was investigated. The results showed that the shape and position of the upper and lower boundaries of the friction change continuously with the change in the posture of the cutter under vibration. Using the above boundary identification method, the

time-varying friction contact interface of the square shoulder milling cutter could be revealed.

(3) The friction speed of the surface element in the friction contact area was studied. The instantaneous friction energy consumption and the friction coefficient model were both developed. According to the equivalent stress decomposition, the calculation methods for the instantaneous normal stress and instantaneous friction of the area element were proposed. The results showed that the friction velocity, friction energy consumption, friction coefficient and friction force in the friction area of the minor flank of the cutter tooth changes with time. The impact between the milling cutter and the workpiece would change the instantaneous friction speed of the surface element of the minor flank of the cutter tooth. Besides, affected by the cutting temperature and stress, the variation in the friction energy consumption and friction coefficient in the friction contact area of the minor flank of the cutter tooth show unsteady characteristics.

(4) The identification method for the variation in the friction variables of the cutter tooth was proposed and verified by experiments. Using the friction characteristic variable model of the minor flank of the cutter tooth, the variability of the friction process of the flank of the cutter tooth could be revealed. The validation experiments results showed that the average correlation coefficient between the cumulative energy consumption boundary and the measured wear boundary was 0.86, which proved the accuracy of the proposed method.

**Author Contributions:** Conceptualization, B.J. and W.L.; methodology, B.J. and W.L.; software, W.L. and L.F.; validation, P.Z. and M.S.; investigation, W.L. and L.F.; resources, P.Z.; data curation, W.L. and M.S.; writing—original draft preparation, W.L.; writing—review and editing, W.L. and P.Z.; supervision, B.J. and P.Z.; project administration, B.J.; funding acquisition, B.J. All authors have read and agreed to the published version of the manuscript.

**Funding:** This research was funded by the National Nature Science Foundation of China, 51875145 and the Nature Science Foundation of Heilongjiang Province of China, ZD2020E008.

**Institutional Review Board Statement:** Not applicable.

**Informed Consent Statement:** Not applicable.

**Data Availability Statement:** No data reported for this study, we have chosen to exclude this claim.

**Conflicts of Interest:** The authors declare no conflict of interest.

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
