# Peer review of "Identification of Friction Behavior Variation in the Minor Flank of Square Shoulder Milling Cutters under Vibration"

_applsci, doi:10.3390/app12084038_

Round 1

Reviewer 1 Report

The paper « Identification for Friction Behavior Variation of the Minor Flank of Square Shoulder Milling Cutter under Vibration » aims to reveal the relationship between machined surface quality, friction and tool wear condition. The works respond to a technological and scientific challenges for controlling the quality and integrity of surfaces after machining.

However, before the publication of the paper, some clarification is needed.

- a more detailed presentation of the experimental procedure is necessary to understand the work. (Test protocol, machine, sensors and measurements performed, ...)

- the treatment carried out on the experimental measurements for the validation of the various models must be presented, in particular the friction.

- Is the vibration measured during the tests?

- what are the software used for the implementation of the developed models ?

- are the friction coefficients and the evolution of wear measured? if yes, how?

- how are the images of figure 4 obtained ?

- The simulations of figure 6 must be detailed is explained in the text

- How does the wear of the milling tool modify the friction? Have the different phases of the wear evolution as a function of time been identified? Is this taken into account in the developed model?

- Please quote table 1 in the text,

- please correct the legend in figure 15

Reviewer 2 Report

Dear editor/author,

I specified my reviews about manuscript as below;

-What was the reason for choosing a three-edged cutting tool? Also, is the recommended system applicable to solide tools? Couldn't optimum test results be obtained by choosing Solide tools?

- The experment expression in Table 3 should be corrected.

-Figures for experimental and simulation should be edited as a and b in Fig.4. This is true for all figures.

- Was the test repeated for experimental and simulation results? Were the results obtained by performing a single experiment under the same conditions?

-What is the material used for the experiments? It would be better if figures of wear are given in comparative form.

Kind regards

Round 2

Reviewer 2 Report

Manuscript can be accepted for publishing.

Kind regards.